# Impact and Tolerance of Immunosuppressive Treatments in Patients Living with HIV with Inflammatory or Autoimmune Diseases

**DOI:** 10.3390/microorganisms10101891

**Published:** 2022-09-23

**Authors:** Zélie Guitton, Nathalie Viget, Laure Surgers, Antoine Cheret, Clotilde Fontier, Laurène Deconinck, Pierre Bataille, Agnès Meybeck, Hélène Bazus, Olivier Robineau

**Affiliations:** 1Infectious Disease Department, Dr Schaffner Hospital, F-62300 Lens, France; 2Service Universitaire des Maladies Infectieuses, CH de Tourcoing, F-59210 Tourcoing, France; 3INSERM, Institut Pierre Louis d’Epidémiologie et de Santé Publique, GHU AP-HP, Sorbonne Univsersité, Service des Maladies Infectieuses et Tropicales, F-75012 Paris, France; 4AP-HP, Service de Médecine Interne et d’Immunologie Clinique, Hôpital Bicêtre, Le Kremlin-Bicêtre, plateforme de médecine ambulatoire CHU de Guadeloupe, UFR Médecine, Université des Antille, F-94270 Kremlin-Bicêtre, France; 5Infectious Disease Department, Valenciennes Hospital, F-59300 Valenciennes, France; 6Infectious Disease Department, Raymond Poincarré Hospital, APHP, F-92380 Garches, France; 7Infectious Disease Department, Boulogne sur Mer Hospital, F-62200 Boulogne sur Mer, France; 8Univ. Lille, CHU Lille, ULR 2694 - METRICS: Évaluation des Technologies de santé et des Pratiques Médicales, F-59000 Lille, France

**Keywords:** HIV, immunosuppressive treatment, inflammatory diseases, auto-immune disorder, methotrexate, anti-TNFα

## Abstract

Background: Patients living with HIV (PLWHIV) can develop autoimmune diseases (AD) needing immunosuppressive treatments (IST). This study aims to describe the impact of IST in PLWHIV. Methods: This was a multicentric retrospective observational study in six HIV referral centers on PLWHIV under IST for AD. Demographic factors, viral co-infections, immunovirological status before and under IST, infectious events, and their descriptions were collected and described focusing on infectious events, immunovirological variations, and IST effectiveness. Results: 9480 PLWHIV were screened for inclusion. Among them, 138 (1.5%) had a history of auto-immune disease, among which 32 (23%) received IST. There was mainly spondyloarthropathy (28%) and the most commonly used IST was methotrexate. The median follow-up under IST was 3.8 years (2.7; 5.9). There were 15 infectious events (0.5 events/individuals) concerning nine patients. At the last medical follow-up, 81% of these were in remission of their AD. Under IST, there was an increase in CD4 during follow-up (629 vs. 827 CD4/mm^3^, *p* = 0.04). No HIV virological failure was noted. Conclusions: This study supports a growing evidence base that IST can be used safely and effectively in PLWHIV with careful monitoring.

## 1. Introduction

As in the general population, patients living with HIV (PLWHIV) can develop autoimmune and inflammatory diseases (AD). The incidence and prevalence of severe AD were considerably higher before the area of highly active antiretroviral therapy (HAART), due to the immunity modification associated with viral replication [1]. A recent Canadian study found that ADs are more prevalent in PLWHIV with an adjusted hazard ratio of 2.40 (95%CI [2.10–2.75]). The strongest associations were seen for hematological disorders, followed by ankylosing spondylitis and inflammatory bowel disease [2]. However, there is a lack of data in PLWHIV on the evolution of AD under immunosuppressive treatment (IST).

Moreover, IST side effects are not well-known in PLWHIV. ISTs are known to increase the rate of serious and opportunistic infections, and they must be employed with great caution in immunosuppressed populations [3]. Clinicians are often reluctant to use them in PLWHIV. Consequently, IST in the HIV population is poorly described in the literature. Regarding disease-modifying antirheumatic drugs (DMARDs), a few case reports described a favorable evolution in HIV patients treated with azathioprine [4,5,6], mycophenolate mofetil (MMF) [7,8], and methotrexate (MTX) [9], and the safety and effectiveness of this medication appeared non-threatening. Concerning biological therapy, such as anti-Tumor Necrosis Factor-alpha medication (anti-TNFα), some studies [10,11,12,13,14] and reviews [15,16] have described no significant clinical adverse effect and a clinical improvement.

Therefore, it appears necessary to evaluate the infectious complications of IST in this specific population in a larger cohort of patients. Our study aimed to assess the proportion of infectious events in PLWHIV treated with IST for AD or inflammatory disease and to evaluate their effectiveness and the variations in immunovirological parameters under treatment.

## 2. Materials and Methods

### 2.1. Ethical Procedures

The two databases from which the data were extracted (Nadis and Diam, consisting of electronic medical records) were declared to the French national commission of liberty and informatics (CNIL), as numbers 1251720 and 462278, respectively. Patients signed written consent for the use of their data for research purposes.

### 2.2. Population Study

We designed a retrospective multicentric study on all PLWHIV aged over 18 years who received follow-up in six HIV referral centers in France (Saint-Antoine (Paris), Kremlin Bicêtre, Tourcoing, Garches, Valenciennes, Boulogne-sur-Mer) between March 2016 and March 2018. We screened patients by using the following CIM-10 diagnosis classification terms associated with AD: “erythematosus lupus”, “dermatomyositis”, “Crohn disease”, “ulcerative colitis”, “spondyloarthritis”, “rheumatoid arthritis”, “autoimmune hepatitis”, “sarcoidosis”, “multiple sclerosis”, “systemic sclerosis”, “vasculitis”, “Still disease” in the HIV population receiving follow-up in each center. We decided to exclude diseases known to be associated with the immunovirological status of the patient: thrombocytopenic purpura and cutaneous psoriasis. Medical records of each preselected patient were retrospectively reviewed to avoid errors in CIM-10 coding. Exclusion criteria were a history of transplantation and neoplasia. We included in our final analysis PLWHIH who had a history of at least one year of IST for AD. IST considered were MMF, sulfasalazine, ciclosporin, MTX, anti-TNFα (infliximab, etanercept, certolizumab, adalimumab), abatacept, anakinra, tocilizumab, and rituximab.

Events of interest were all infectious events occurring after IST introduction, excluding sexually transmitted diseases. Infectious events were collected by chart review and were defined as an event that was microbiologically documented or clinically suggestive of an infectious etiology.

### 2.3. Clinical and Biological Variables

All variables were collected in a case report form. Sociodemographic characteristics and comorbidities were collected (diabetes, Chronic Obstructive Pulmonary Disease (COPD), chronic kidney disease (CKD), cirrhosis, hepatitis). HIV disease was described by the type of transmission, date of diagnosis, HIV stage (Center for Diseases Control (CDC) stage). CD4 rate and viral load collected were those that were the closest to the beginning of the HIV follow-up, to the time IST was started, and to the last visit to the centers. Treatment history was also described. The AD was described by the year of diagnosis, the type of IST with the date of its introduction and discontinuation, the association with corticosteroids, and the tuberculosis screening assessment before treatment introduction. Infectious events under IST were described (type, severity, and outcome).

### 2.4. Statistical Analyses

Qualitative parameters were expressed in terms of frequency and percentage; Gaussian numerical parameters were expressed in terms of mean and standard deviation, and non-Gaussian numerical parameters in terms of median and interquartile interval. CD4 rate at the beginning of IST was compared with the CD4 rate at the end of follow-up using *t*-test. Statistical analyses were performed using R software (version 4.0.5., Vienna, Austria).

## 3. Results

A total of 9480 PLWHIV were screened for inclusion. Among these, 138 (1.5%) patients had a history of AD, among which 32 (23%) received IST. Patients’ baseline characteristics at the time of HIV diagnosis are summarized in Table 1. Briefly, patients had a median age of 34.5 years (26–41), and 44% (14/32) were females. There were no patients with CKD or COPD, and 6% (2/32) presented an association with diabetes. Concerning HIV history, the median duration of follow-up was 9.8 years (4–17.5).

Thirteen patients were diagnosed with HIV while they already had a history of AD. Among these, one was infected while he was under IST. Three patients were diagnosed with HIV during explorations made for the initiation of IST. Sixty percent of patients (19/32) were diagnosed with AD after HIV diagnosis. 

The description of AD and IST are shown in Table 2. There were mainly spondyloarthropathies (9 patients, 28%), associated or not with psoriasis, followed by chronic inflammatory bowel diseases (7 patients, 22%) and systemic lupus erythematosus (2 patients, 6%). The most frequently prescribed IST was methotrexate (12 patients, 38%) followed by azathioprine, sulfasalazine, and infliximab (5 patients, 16% for each). Corticosteroids were associated with IST for 18 patients (56%). Nine patients received at least two successive IST. The median follow-up under IST was 3.8 years (2.7–5.9). At the last medical follow-up, 26 patients (81%) were in remission of AD.

Among these patients under IST, 15 infectious events occurred (0.5 events/individuals). They concerned nine patients (28%). The median delay of infectious event apparition was 1.2 years (min–max: 01–13). Among these, 6 were declared during the first year of IST. Hospitalization was required for 6 events (20%) and concerned 5 individuals. Pulmonary infection was the most frequent type of infection, followed by upper respiratory tract infection. Under IST, there was no viral reactivation. No death was associated with infectious events (Table 3). IST was discontinued for one patient due to a long remission. Under IST, there was an increase in CD4 between the beginning of the treatment and the end of the follow-up (629 vs. 827 CD4/mm^3^, *p* = 0.04).

## 4. Discussion

In this study, AD prevalence in PLHIV is about 1.5%, with a predominance of spondyloarthropathy, associated or not with psoriasis, followed by chronic inflammatory bowel disease. Infectious events under IST concerned more than a quarter of patients and were mainly pulmonary infections. Under IST, there was no viral reactivation, and we noted a significant increase in CD4 rate.

In the literature, AD prevalence varies between 0.69% and 9.6% [2,17,18,19]. This wide range might be explained by the location of the studies and bias due to the fact that most of them were monocentric. In this study, inflammatory bowel disease is more prevalent than in other series. This might be due to the location of the center involved, mainly located in the north of France, where this disease is more prevalent [20]. Concerning diseases that might need IST, this work confirmed a similar prevalence in PLWHIV when compared to the general population, as already suggested [2].

ISTs are associated with a high rate of infection in the overall population. A meta-analysis including 78 studies with 24 996 patients highlighted that anti-TNF drug use is associated with an increase in the occurrence of infections (+20%), serious infections (+40%), and tuberculosis (+250%) [21]. Methotrexate is associated with an increased risk of infection in rheumatoid arthritis (RR: 1.25; 95% CI [1.01–1.56]; *p* = 0.04) in another meta-analysis including 1146 patients [22]. Interestingly, in the PLWHIV population, this rate seems to be similar. In the present study, there were 15 infectious events, with no death, and one patient required an intensive care unit. Although DMARDs are the usual first-line IST, only a few case reports described their use in PLWHIV. Case reports using azathioprine [4,5,6], mycophenolate mofetil (MMF) [7,8], and methotrexate (MTX) [9] described favorable evolution without infectious complication. Concerning anti-TNF α, reviews including 27 patients [15] and 37 treatment episodes [16], and a retrospective study describing 26 patients treated with anti-TNF α [13] found that the rate of serious infections may be comparable to the rates observed in the registry databases. One study including systemic therapy and biological drugs in 33 psoriatic PLWHIV confirmed this assumption [10]. Among the five PLWHIV with AIDS under this type of treatment, four presented infectious events. These data suggest that this kind of treatment should be used in patients immunologically controlled at the start of these treatments. However anti-TNF α is also sometimes used as immune reconstitution inflammatory syndrome (IRIS) medication in immunocompromised patients without major infectious events [23,24,25].

Interestingly, CD4 cell count rose under IST. There is a lack of data on CD4 evolution under IST in immunocompetent patients, except for rituximab which induces CD4 T cell depletion [26]. Methotrexate and anti-TNF α does not seem to induce CD4 count modification [27,28]. This rise in the rate is probably related to the natural evolution under antiretroviral treatment. IST does not seem to affect this positive evolution under HAART.

There are no guidelines from medical societies on how and when to start IST in PLWHIV. A recent retrospective monocentric study including 77 patients treated with immunomodulatory or biologic therapy for AD or oncology proposed recommendations for the use of immunomodulatory drugs in PLWHIV [14]. They suggested assessing vaccination and to search for latent infection (tuberculosis, hepatitis). During follow-up, they suggest controlling the plasma HIV RNA level and CD4 count every three months during the first year of therapy. It appears that IST can be started using the same criteria as in the general population when the CD4 rate is restored.

This study supports a growing evidence base that IST can be used safely and effectively in PLWHIV with careful monitoring, even though these individuals were not included in the initial clinical trials. These data from a large network of French hospitals, add to previously published reports from other more limited centers, mostly in the U.S. The most important next steps for investigation might include prospective studies from PLWHIV under IST including individuals with less robust immune reconstitution, to better ascertain the additional risk of infectious complications.

## Figures and Tables

**Table 1 microorganisms-10-01891-t001:** Baseline characteristics.

Variables	Value
Age (year) (median, IQR)	34.5 (26–41)
Gender Female (n, %)	14 (43.8)
Coinfections	
- HCV (n, %) - HBV (n, %)	0 (0) 1 (3.1)
Comorbidity	
- Diabetes (n, %) - Cirrhosis (n, %) - CKD (n, %) - COPD (n, %)	2 (6.25) 0 (0) 0 (0) 0 (0)
Contamination mode	
- Heterosexual (n, %) - MSM (n, %) - Unknown (n, %)	15 (46.9) 15 (46.9) 2 (6)
Immunovirologic status (at diagnosis)	
- CD4 rate (cell/mm^3^) (median, IQR) - % CD4 (median, IQR) - nadir CD4 (cell/mm^3^) (median, IQR) - nadir CD4 % - viral load (log) (median, IQR)	582 (280–848) 32 (24.0–40.8) 258 (130–428) 31 (21.0;35.0) 4.46 (2.88–5.09)
HLA	
- B27 (n, %) - B57 (n, %)	5 (15.6) 2 (6.25)

**Table 2 microorganisms-10-01891-t002:** Description of patient auto-immune disease, immunosuppressive treatment and outcome.

Autoimmune disease	
- autoimmune hepatitis (n, %) - erythematosus lupus (n, %) - Crohn’s disease (n, %) - Still disease (n, %) - rheumatoid arthritis (n, %) - psoriatic arthritis (n, %) - Ulcerative colitis (n, %) - Sarcoidosis (n, %) - Spondyloarthritis (n, %) - Vasculitis (n, %)	3 (9) 2 (6) 4 (12.5) 2 (6) 4 (12.5) 3 (9) 3 (9) 1 (3) 9 (28) 1 (3)
Immunosuppressive treatment	
- Azathioprine (n, %) - Methotrexate (n, %) - Mycophenolate mofetil (n, %) - Sulfasalazine (n, %) - Ciclosporin (n,%) - Anakinra (n, %) - Abatacept (n, %) - Rituximab (n, %) - Tocilizumab (n, %) - Infliximab (n, %) - Etanercept (n, %) - Adalimumab (n, %) - Certolizumab (n, %)	5 (16) 12 (38) 4 (13) 5 (16) 1 (3) 2 (6) 1 (3) 1 (3) 1 (3) 5 (16) 4 (13) 4 (13) 1 (3)
Infectious events under IST	
- Pneumonia (n, %) - Upper respiratory tract infectious (n, %) - Cutaneous infection (n, %) - Digestive (n, %)	6 (40) 4 (27) 3 (20) 2 (13)
Hospitalization for an infectious event (n; %)	6 (40)

**Table 3 microorganisms-10-01891-t003:** Description of the infectious events.

Patient	Gender	Age	AD Type	Type of Infection	Type of IST	Delay between IST Initiation and Infectious Event (Year)	Management	Death
1	F	57	Autoimmune hepatitis	dermohypoderma	Mycophenolate mofetil	9.1	hospitalization	no
2	M	64	Rheumatoid Arthritis	tuberculosis reactivation	Methotrexate	0.1	hospitalization	no
2	M	64	Rheumatoid Arthritis	bacterial pneumonia	Abatacept	0.9	hospitalization	no
2	M	64	Rheumatoid Arthritis	salmonella colitis	Abatacept	2.2	outpatient	no
2	M	64	Rheumatoid Arthritis	bacterial pneumonia	Abatacept	3.5	outpatient	no
3	F	68	Crohn disease	bacterial otitis	Infliximab	1.9	outpatient	no
3	F	68	Crohn disease	oral candidiasis	Infliximab	2.5	outpatient	no
4	F	76	Spondyloarthritis	cutaneous abscess	Etanercept	12.0	outpatient	no
5	F	51	Still disease	bacterial pneumonia	Ciclosporin	0.1	hospitalization	no
5	F	50	Still disease	oral candidiasis	Ciclosporin	1.2	outpatient	no
6	M	47	Vasculitis	clostridium colitis	Rituximab	0.1	hospitalization	no
6	M	47	Vasculitis	dental abscess	Rituximab	0.2	outpatient	no
7	M	52	Spondyloarthritis	bacterial pneumonia	Etanercept	1.2	outpatient	no
8	M	32	Spondyloarthritis	zoster	Adalimumab	13.0	outpatient	no
9	F	74	Lupus	legionella pneumonia	Mycophenolate mofetil	0.3	hospitalization	no

AD: Auto-immune disease; IST: immunosuppressive treatment.

## Data Availability

Data are available upon reasonable request to the corresponding author.

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
