# Peer review of "Impact and Tolerance of Immunosuppressive Treatments in Patients Living with HIV with Inflammatory or Autoimmune Diseases"

_microorganisms, 2022, doi:10.3390/microorganisms10101891_

Round 1

Reviewer 1 Report

Dear Authors,

The manuscript entitled “Impact and tolerance of immunosuppressive treatments in patients living with HIV with inflammatory or autoimmune diseases” by Guitton et al. is an interesting paper describing and adding supporting new data to the important issue regarding the effects of immunosuppressive treatments (IST) in people living with HIV (PLWHIV) affected by autoimmune diseases. The Authors describe a multicentric study aimed at evaluating the occurrence of infections in PLWHIV treated with IST for autoimmune or inflammatory diseases, the effectiveness of IST and the variations in immunovirological parameters under treatment. The article is generally clear and well written, the results are well presented and discussed, and the findings give a potential improvement to the knowledge in this field. Therefore, after a few English editing and the revision of the following minor points, I would suggest that the manuscript is suitable for publication in Microorganisms.

Minor points: 

Section 3. Results, Page 3, Line 112: in the sentence “Briefly, patients had a median age of 34.5 years [26-29]...”, Authors should please check if the value of median age, or the values from which it was calculated, are correctly reported. The same values are also reported in Table 1. Baseline characteristics, please check. 

Section 3. Results, Page 4, Lines 125-26: in the sentence “The most frequently prescribed IST was methotrexate (12 patients, 38%) followed by azathioprine and infliximab (5 patients, 16% for each)”, there seems to be an inconsistency with the values reported in Table 2: in fact, as reported in Table 2 at page 5, the most frequently prescribed immunosuppressive treatments seem to be Methotrexate and Azathioprine with the same percentage, while Infliximab seems to be like Sulfasalazine, please check.  

Section 4. Discussion, Page 7, Line 164: in the sentence “In the present study, there were 25 infectious events...”, Authors should please check if the value of infectious events is correctly reported. 

Section 4. Discussion, Page 7, Line 165:please spell out the acronym ICU.  

Some typos have to be corrected all over the text (i.e., lines 83: PLWHIW in PLWHIV, Table 2: Upper respiratory infectious in Upper respiratory infections, Table 2: Immunosupressive treatment in Immunosuppressive treatment, Table 3: about patients 4 and 6, cutaneous abcess in abscess, Table 3: please correctly report the punctuation of the years’ values. 

Author Response

Dear Editor, Dear Reviewers,

 Thank you for your careful review of our manuscript. In this new version, we hope to have answered all your requests. Furthermore, the manuscript has been reviewed by an English native making some sentences clearer. Please find attached a point-by-point answer to your remarks.

Sincerely,

Zelie Guitton

Comments and Suggestions for Authors

Dear Authors,

The manuscript entitled “Impact and tolerance of immunosuppressive treatments in patients living with HIV with inflammatory or autoimmune diseases” by Guitton et al. is an interesting paper describing and adding supporting new data to the important issue regarding the effects of immunosuppressive treatments (IST) in people living with HIV (PLWHIV) affected by autoimmune diseases. The Authors describe a multicentric study aimed at evaluating the occurrence of infections in PLWHIV treated with IST for autoimmune or inflammatory diseases, the effectiveness of IST and the variations in immunovirological parameters under treatment. The article is generally clear and well written, the results are well presented and discussed, and the findings give a potential improvement to the knowledge in this field. Therefore, after a few English editing and the revision of the following minor points, I would suggest that the manuscript is suitable for publication in Microorganisms.

Minor points: 

Point 1. Section 3. Results, Page 3, Line 112: in the sentence “Briefly, patients had a median age of 34.5 years [26-29]...”, Authors should please check if the value of median age, or the values from which it was calculated, are correctly reported. The same values are also reported in “Table 1. Baseline characteristics”, please check. 

Response 1. Thank you for this remark, There were typos error. We reviewed the whole manuscript. Concerning this point, median age was 34.5 [26-41]

Point 2. Section 3. Results, Page 4, Lines 125-26: in the sentence “The most frequently prescribed IST was methotrexate (12 patients, 38%) followed by azathioprine and infliximab (5 patients, 16% for each)”, there seems to be an inconsistency with the values reported in Table 2: in fact, as reported in Table 2 at page 5, the most frequently prescribed immunosuppressive treatments seem to be Methotrexate and Azathioprine with the same percentage, while Infliximab seems to be like Sulfasalazine, please check.  

Response 2. There was a mistake in Table 2 : Azathioprine was prescribed for 5 patients (16%) we changed the value in Table 2. We also changed the text as follow: “The most frequently prescribed IST was methotrexate (12 patients, 38%) followed by azathioprine, sulfasalazine, and infliximab (5 patients, 16% for each).”

Point 3. Section 4. Discussion, Page 7, Line 164: in the sentence “In the present study, there were 25 infectious events...”, Authors should please check if the value of infectious events is correctly reported. 

Response 3 : There were typo error. Sorry for that. We changed the text as follow:“. In the present study, there were 15 infectious events, with no death, and one patient required Intensive Care Units »

Point 4. Section 4. Discussion, Page 7, Line 165: please spell out the acronym ICU.

Response 4 : We replaced this acronym by Intesive Care Unit

Point 5. Some typos have to be corrected all over the text (i.e., lines 83: PLWHIW in PLWHIV, Table 2: Upper respiratory infectious in Upper respiratory infections, Table 2: Immunosupressive treatment in Immunosuppressive treatment, Table 3: about patients 4 and 6, cutaneous abcess in abscess, Table 3: please correctly report the punctuation of the years’ values. 

Response 5 : We did all that changes. Thank you for this careful review.

Reviewer 2 Report

This manuscript describes a study on the impact of immunosupressive treatments on patients living with HIV who developed autoimmune diseases. The study shows around 1.5% of patients develop autoimmune diseases, and during follow-up CD4 cell number increased under IST. The methods are and results are clearly described and the discussion nicely compares and supports the notion that IST can be administered safely.

Author Response

Dear Editor, Dear Reviewers,

 Thank you for your careful review of our manuscript. In this new version, we hope to have answered all your requests. Furthermore, the manuscript has been reviewed by an English native making some sentences clearer. Please find attached a point-by-point answer to your remarks.

Sincerely,

Zelie Guitton

This manuscript describes a study on the impact of immunosupressive treatments on patients living with HIV who developed autoimmune diseases. The study shows around 1.5% of patients develop autoimmune diseases, and during follow-up CD4 cell number increased under IST. The methods are and results are clearly described and the discussion nicely compares and supports the notion that IST can be administered safely.

Thank you.